# InfoMat: Leveraging Information Theory to Visualize and Understand Sequential Data [note 1]

**DOI:** 10.3390/e27040357

**Published:** 2025-03-28

**Authors:** Dor Tsur, Haim Permuter

**Affiliations:** School of Electrical and Computer Engineering, Ben-Gurion University of the Negev, Be’er Sheva 8410501, Israel; haimp@bgu.ac.il

**Keywords:** data analysis, data visualization, directed information, information matrix, information conservation, mutual information, transfer entropy

## Abstract

Despite the widespread use of information measures in analyzing probabilistic systems, effective visualization tools for understanding complex dependencies in sequential data are scarce. In this work, we introduce the information matrix (InfoMat), a novel and intuitive matrix representation of information transfer in sequential systems. InfoMat provides a structured visual perspective on mutual information decompositions, enabling the discovery of new relationships between sequential information measures and enhancing interpretability in time series data analytics. We demonstrate how InfoMat captures key sequential information measures, such as directed information and transfer entropy. To facilitate its application in real-world datasets, we propose both an efficient Gaussian mutual information estimator and a neural InfoMat estimator based on masked autoregressive flows to model more complex dependencies. These estimators make InfoMat a valuable tool for uncovering hidden patterns in data analytics applications, encompassing neuroscience, finance, communication systems, and machine learning. We further illustrate the utility of InfoMat in visualizing information flow in real-world sequential physiological data analysis and in visualizing information flow in communication channels under various coding schemes. By mapping visual patterns in InfoMat to various modes of dependence structures, we provide a data-driven framework for analyzing causal relationships and temporal interactions. InfoMat thus serves as both a theoretical and empirical tool for data-driven decision making, bridging the gap between information theory and applied data analytics.

## 1. Introduction

Information theory plays a key role in the analysis of dynamics in stochastic systems [1,2,3,4]. Through the lens of information theory, one can study the temporal evolution of dependence in a sequential system, which is often interpreted as the exchange of information between its interacting components. For example, in communication channels, directed information quantifies the information flow from the encoder to the channel [5,6]. Consequently, the feedback capacity of communication channels, which is characterized by the optimization of average directed information, can be interpreted as the maximum amount of information flow from the transmitter to the channel [7]. Another pertinent example is neuroscience [8], where information theory is widely used for the analysis and processing of collected data. For example, transfer entropy [9] is used to infer functional connectivity and to analyze information flow from recorded data, such as EEG and fMRI [10]. Beyond communications and neuroscience, information theory was shown useful for various fields of sequential analysis, encompassing control [11,12,13], reinforcement learning [14,15], causal inference [16,17,18], and various machine learning tasks [19,20,21].

In contrast to its wide use to infer and quantify relations in time series data, information theory fails to offer simple visualization tools to demonstrate its merits, and existing visualization techniques are not applicable to such settings. A common example of dependence visualization is the cross-correlation matrix [22]. For a given random vector pair $\left(X,Y\right)\in R^{d_{x}+d_{y}}$, the correlation matrix is an $R_{d_{x}\times d_{y}}$-valued matrix, whose (i,j)th entry is the correlation between $\left(X_{i},Y_{j}\right)$. While being a powerful tool for simple exploratory analysis, the performance of the correlation matrix heavily relies on the data domain and structure of the joint distribution. Specifically, it provides a complete characterization of dependence only when the vectors’ entries are univariate Gaussian sequences with linear relations. Furthermore, the correlation matrix does not generalize to the conditional setting, which is crucial for visualization in time series analysis.

A common visualization of information theoretic relationships is the Venn diagram [4], which serves as a visualization of mutual information, through decomposition into joint and conditional entropies. Finally, the information diagram [23], which is based on the Taylor diagram [24], explores the interplay between entropies and mutual information in a geometric fashion by mapping dependence into an angle between the marginal entropy vectors. However, this method is limited to a random pair. Beyond the visualization of information measures, information theory is widely used to evaluate other visualization and rendering techniques [25]. In this work, we attempt to close this gap, and propose a new visualization method.

### Contributions

In this work, we propose the information matrix (InfoMat), a novel visualization tool of the information transfer in dynamic stochastic systems. Given two stochastic sequences of length *m*, the InfoMat arranges the conditional mutual information terms that describe the evolution of dependence in an m×m matrix. By arranging the $m^{2}$ conditional information terms that describe the temporal processes’ evolution in an m×m matrix, various dependence patterns emerge in the InfoMat though visual patterns. We show how the InfoMat captures popular sequential information measures such as the directed information and transfer entropy through linear operations. We then demonstrate the InfoMat utilities for both theoretical and practical methods.

For theoretical purposes, the InfoMat provides a visual representation of existing information theoretic conservation laws and decompositions, while revealing new relationships. The relationships are proven by characterizing different subsets of the matrix with corresponding information measures. Additionally, the InfoMat serves as a practical visualization tool for arbitrary sequential data. Using a heatmap representation, we can link various dependence structures in the data with visual patterns. We propose a Gaussian mutual information estimator of the InfoMat that relies on the calculation of sample covariance matrices and analyze its theoretical guarantees, while empirically demonstrating the power of the resulting visualization tool. When a Gaussian estimator is insufficient, we develop a neural InfoMat estimator, which is based on masked normalizing flows (MAFs), which expand the class of distributions captured by the InfoMat. We demonstrate the estimated InfoMat to visualize the power of optimal coding schemes in communication channels with memory and for the visualization of information flows in real-world datasets.

The rest of this paper is organized as follows. Section 2 presents required background and preliminaries, followed by Section 3, which presents the InfoMat and demonstrates its capabilities for the visualization of theoretical information-theoretic conservation laws. Then, Section 4 discusses the Gaussian estimator of the InfoMat, while Section 5 explores neural estimation. Finally, Section 6 demonstrates the utility of the estimated InfoMat for visualizing dependence structures in sequential data, and Section 7 provides concluding remarks and discusses future work.

## 2. Background and Preliminaries

### 2.1. Notation

Sets are denoted by calligraphic letters, e.g., X. When X is finite, we use |X| for its cardinality. For any n∈N, $X^{n}$ is the *n*-fold Cartesian product of X, while $x^{n}=\left(x_{1},\ldots ,x_{n}\right)$ denotes an element of $X^{n}$. For i,j∈Z with i≤j, we use the shorthand $x_{i}^{j}≜\left(x_{i},\ldots ,x_{j}\right)$; the subscript is omitted when i=1. Expectations are denoted by E. When X is countable, we use *p* for the PMF associated with the probability measure *P*. Random variables are denoted by upper-case letters, e.g., *X*. The Kullback–Leibler (KL) divergence between *P* and *Q*, with P≪Q, is $D_{\mathrm{KL}}\left(P∥Q\right)≜E_{P}\left[\log\frac{dP}{dQ}\right]$. The mutual information between $\left(X,Y\right)∼P_{XY}$ is $I\left(X;Y\right)≜D_{\mathrm{KL}}\left(P_{XY}∥P_{X}⊗P_{Y}\right)$, where $P_{X}$ and $P_{Y}$ are the marginals of $P_{XY}$. The entropy of a discrete random variable X∼P is $H\left(X\right)≜-E\left[\log p(X)\right]$.

### 2.2. Sequential Information Measures

Consider a pair of jointly distributed sequences of length *m*, $\left(X^{m},Y^{m}\right)∼P_{X^{m},Y^{m}}$. The causal nature of sequential communication systems impedes mutual information from properly describing sequential information flows, as it decomposes into non-causal conditional mutual information terms, i.e., [26]$$I\left(X^{m};Y^{m}\right)=\sum_{i=1}^{m}I\left(X_{i};Y^{m}|X^{i-1}\right),$$
To this end, several causal adaptations of mutual information to time series data were developed in the literature. The first is directed information [5], which was originally developed to characterize information rates in communication channels with feedback. The directed information *from* $X^{m}$ *to* $Y^{m}$ is given by(1)$$I\left(X^{m}\to Y^{m}\right)≜\sum_{i=1}^{m}I\left(X^{i};Y_{i}|Y^{i-1}\right),$$
where the directed information in the opposite direction $I(Y^{m}\to X^{m})$ is defined symmetrically. The second information measure, which gained popularity in neuroscience and physics is transfer entropy [9,27,28]. For parameters (m,k,l), the transfer entropy is given by(2)$$T_{m}^{X\to Y}\left(k,l\right)≜I\left(X_{m-k}^{m-1};Y_{m}|Y_{m-l}^{m-1}\right),$$
Transfer entropy and directed information follow various decompositions and information conservation laws. We will further discuss them through the lenses of the proposed InfoMat in the upcoming section.

In physics, time series measures—such as approximate entropy [29], sample entropy [30], permutation entropy [31], and dispersion entropy [32]—are widely used to quantify the local complexity and irregularity of sequential data. While these scalar metrics provide valuable insights into local dynamical properties, they are conceptually distinct from directed information and transfer entropy, as they are not based on quantification through conditional mutual information.

### 2.3. Information Decomposition and Conservation

In stochastic systems with memory, the temporal evolution of the dependence between the system’s elements can be viewed through the lenses of information exchange. This exchange of information can be captured within conversation laws, which quantify the total amount of information flow in a given system. Specifically, for a system with *m* time steps and two sources $\left(X^{m},Y^{m}\right)∼P_{X^{m},Y^{m}}$, Massey [33] proved the following law of information conservation(3)$$I\left(X^{m};Y^{m}\right)=I\left(X^{m}\to Y^{m}\right)+I\left(D\circ Y^{m}\to X^{m}\right)$$
where $\left(D^{k}\circ X^{m}\right)$ is a left concatenation of *k* ‘dummy’ deterministic symbols with $X_{k+1}^{m}$ and $\left(D\circ X^{m}\right)=\left(D^{1}\circ X^{m}\right)$. The authors of [34] propose a modification of (Equation 3) that distinguishes between past and present effects, given by(4)$$\begin{array}{c} I\left(X^{m};Y^{m}\right)=I\left(D\circ X^{m}\to Y^{m}\right)+I\left(D\circ Y^{m}\to X^{m}\right)+I_{\mathrm{inst}}\left(X^{m},Y^{m}\right), \end{array}$$
where $I_{\mathrm{inst}}\left(X^{m},Y^{m}\right)≜\sum_{i=1}^{n}I\left(X_{i};Y_{i}|X^{i-1},Y^{i-1}\right)$ is the *instantaneous* information, which measures the symmetric dependence between $X^{m}$ and $Y^{m}$, given a shared history. Information conservation has been shown to be useful for the analysis of causal information flows in neural spike trains and financial data [35,36].

## 3. Information Matrix

Fix m∈N. We are interested in characterizing the interaction between two stochastic sequences (the sequences can be thought of as an *m*-step sample from some underlying joint stochastic process) of length *m* in a visually meaningful manner. Denote the sequences with $\left(X^{m},Y^{m}\right)∼P_{X^{m},Y^{m}}$. Our emphasis is on the information-theoretic description of this interaction, which can be seen as the evolution of dependence between the interacting components over time. This characterization can alternatively be viewed as a transfer of information. The entire dependence structure between $X^{m}$ and $Y^{m}$ is captured by the *m*-fold mutual information, given by the following chain rule [4](5)$$I\left(X^{m};Y^{m}\right)=\sum_{i=1}^{m}\sum_{j=1}^{m}I\left(X_{i};Y_{j}|X^{i-1},Y^{j-1}\right).$$
The above decomposition implies that, along *m* steps, the interaction is characterized with $m^{2}$ conditional mutual information terms. This acts as our motivation for the following definition and this work.

We define the InfoMat as the following m×m matrix:(6)$$I^{X,Y}\in R_{\geq 0}^{m\times m},\;I_{i,j}^{X,Y}≜I\left(X_{i};Y_{j}|X^{i-1},Y^{j-1}\right).$$
The InfoMat captures all the information transfer within the two-user system $\left(X^{m},Y^{m}\right)$ as (Equation 5) implies that$$I\left(X^{m};Y^{m}\right)=\sum_{i,j}I_{i,j}^{X,Y}=1_{m}I^{X,Y}1_{m}^{⊺},$$
where $1_{m}\in R^{m}$ is an *m*-length vector of ones and the inequality follows from the chain rule for mutual information [4].

As we further show, the InfoMat allows for the interpretations of the chain rule (Equation 5), and other information decomposition laws of $I(X^{m};Y^{m})$ as grouping the entries of $I^{X,Y}$ into meaningful subsets that sum up to $I(X^{m};Y^{m})$. These decompositions are often termed information conservation laws [33]. We will show that such laws, whose proofs are usually technical and algebraic, can be easily visualized by coloring subsets of the entries of $I^{X,Y}$.

### 3.1. Visualizing Sequential Information Measures

We begin by demonstrating that the aforementioned sequential information measures can be recovered through the summation of elements of the InfoMat. Directed information (Equation 1) is given by the sum of a triangular sub-matrix, i.e.,(7)$$I(X^{m}\to Y^{m})=1^{T}\left(\begin{array}{cccc} I_{1,1}^{X,Y} & I_{1,2}^{X,Y} & \ldots & I_{1,m}^{X,Y}\\ 0 & I_{2,2}^{X,Y} & \ddots & \vdots \\ \vdots & \ddots & \ddots & I_{m-1,m}^{X,Y}\\ 0 & \ldots & 0 & I_{m,m}^{X,Y} \end{array}\right)1.$$
Due to its direction sensitivity, directed information in each direction is associated with a certain triangular sub-matrix. Specifically, the direction $X^{m}\to Y^{m}$ corresponds to the upper triangular part of $I^{X,Y}$, while the lower triangular part represents the direction $Y^{m}\to X^{m}$.

It is also useful to define the time-delayed directed information, which describes the causal flow of information under a time delay at the transmitting node. For a delay of k<m, the *k*-time-delayed directed information is given by $I\left(D^{k}\circ X^{m}\to Y^{m}\right)≜\sum_{i=1}^{m}I\left(X^{i-k};Y_{i}|Y^{i-1}\right)$. In $I^{X,Y}$, a *k*-delay of directed information corresponds to a right shift of *k* indices in the $X^{m}\to Y^{m}$ direction, and a down shift in the $Y^{m}\to X^{m}$ direction.

Next, the transfer entropy term $T_{i+1}^{X\to Y}\left(i,i\right)=I\left(X^{i};Y_{i+1}|Y^{i}\right)$, which quantifies the causal effect of $X^{i}$ on $Y_{i+1}$ given $Y^{i}$ is given by the sum over a column of length *i* in row i+1. For example, we have(8)$$T_{3}^{X\to Y}\left(2,2\right)=1^{T}\left(\begin{array}{ccccc} 0 & \ldots & I_{1,3}^{X,Y} & \ldots & 0\\ \vdots & 0 & I_{2,3}^{X,Y} & \ddots & \vdots \\ \vdots & \ddots & 0 & \ddots & \vdots \\ \vdots & \ddots & \ddots & 0 & \vdots \\ 0 & \ldots & \ldots & \ldots & 0 \end{array}\right)1.$$
Note that a transfer entropy does not include terms on the InfoMat diagonal. This stresses that transfer entropy focuses on strict past influence on present interaction, which is a key distinction from directed information [37]. The relation between mentioned information measures and the patterns in $I^{X,Y}$ are summarized in Table 1.

### 3.2. Capturing Information Conservation Laws

Having identified the relation between information measures and their patterns in $I^{X,Y}$, we can visualize the aforementioned laws of information conservation [33,34]. Such rules are often derived via algebraic manipulation of information measures and therefore may lack intuition and may fail to provide a deeper understanding of the underlying interaction. Recall that Massey’s information conservation law is given by (Equation 3)(9)$$I\left(X^{m};Y^{m}\right)=I(X^{m}\to Y^{m})+I(D\circ Y^{m}\to X^{m})$$
Following the identification of DI with triangular submatrices of $I^{X,Y}$, (Equation 3) follows by coloring index subsets and summing over each color group.(10)$$\;I\left(X^{m};Y^{m}\right)=1^{T}\left(\begin{array}{cccc} I_{1,1}^{X,Y} & I_{1,2}^{X,Y} & \ldots & I_{1,m}^{X,Y}\\ I_{2,1}^{X,Y} & I_{2,2}^{X,Y} & \ddots & \vdots \\ \vdots & \ddots & \ddots & I_{m-1,m}^{X,Y}\\ I_{m,1}^{X,Y} & \ldots & I_{m,m-1}^{X,Y} & I_{m,m}^{X,Y} \end{array}\right)1.$$
Considering the finer decomposition using instantaneous information (Equation 4), we note that $I_{\mathrm{inst}}\left(X^{n},Y^{n}\right)=\mathrm{Trace}\left(I^{X,Y}\right)$. We can therefore identify this decomposition by excluding the diagonal from the upper sub-triangle.(11)$$\;I\left(X^{m};Y^{m}\right)=1^{T}\left(\begin{array}{cccc} I_{1,1}^{X,Y} & I_{1,2}^{X,Y} & \ldots & I_{1,m}^{X,Y}\\ I_{2,1}^{X,Y} & I_{2,2}^{X,Y} & \ddots & \vdots \\ \vdots & \ddots & \ddots & I_{m-1,m}^{X,Y}\\ I_{m,1}^{X,Y} & \ldots & I_{m,m-1}^{X,Y} & I_{m,m}^{X,Y} \end{array}\right)1.$$
Next, we will leverage the constructed relations between InfoMat patterns and sequential information measures to derive new information-theoretic decompositions and formulas.

### 3.3. Developing New
Information-Theoretic Relations

Beyond the visualization of existing relationships, the simplicity of the InfoMat visualization allows us to develop new meaningful information-theoretic equivalences. We begin with the following proposition that relates the two information measures of interest.

**Proposition** **1**(Transfer entropic decomposition of directed information)**.**
*For $\left(X^{m},Y^{m}\right)∼P_{X^{m},Y^{m}}$ and 1≤k≤m, we have*$$I\left(D^{k}\circ X^{m}\to Y^{m}\right)=\sum_{i=1}^{m-k}T_{i+1}^{X\to Y}\left(i,i\right).$$

**Proof.** We provide the proof for k=1. Recall that directed information corresponds to the upper triangular part of $I^{X,Y}$. We note that $T_{i+1}^{X\to Y}\left(i,i\right)$ corresponds to a column in $I^{X,Y}$ that begins in the (i+1)th column and has length *i*. Thus, we color the triangular part (directed information) as follows$$\left(\begin{array}{ccccc} \; & I_{1,2}^{X,Y} & I_{1,3}^{X,Y} & \ldots & I_{1,m}^{X,Y}\\ & & I_{2,3}^{X,Y} & \ddots & \vdots \\ & & & \ddots & \vdots \\ & & & & I_{m-1,m}^{X,Y} \end{array}\right)$$
The relation then follows by summing over the upper triangular part and dividing the sum into the corresponding rows, i.e.,$$\begin{array}{cc} \; & I\left(D\circ X^{n}\to Y^{n}\right)=T_{2}^{X\to Y}\left(1,1\right)+T_{3}^{X\to Y}\left(2,2\right)+\cdots +T_{m}^{X\to Y}\left(m-1,m-1\right) \end{array}$$
The proof similarly extends to k>1 with similar steps.    □

The proof demonstrates the simplicity of InfoMat, as it boils down to identifying the aforementioned information measures as entry subsets within $I^{X,Y}$. Equipped with Proposition 1, we can provide an information conservation rule in terms of transfer entropy terms.

**Proposition** **2**(Conservation of transfer entropy)**.**
*Let $\left(X^{m},Y^{m}\right)∼P_{X^{m},Y^{m}}$. Then*$$I\left(X^{m};Y^{m}\right)=\sum_{i=1}^{m-1}T_{i+1}^{X\to Y}\left(i,i\right)+T_{i+1}^{Y\to X}\left(i,i\right)+I_{\mathrm{inst}}\left(X^{m},Y^{m}\right).$$

Finally, we propose a recursive decomposition of directed information via transfer entropy.

**Proposition** **3**(Directed information chain rule)**.**
*Let $\left(X^{m},Y^{m}\right)∼P_{X^{m},Y^{m}}$. Then*$$\begin{array}{c} I\left(D^{k}\circ X^{m}\to Y^{m}\right)=I\left(D^{k+1}\circ X^{m}\to Y^{m}\right)+\sum_{i=1}^{m-k}I\left(X_{i-k};Y_{i}|Y^{i-1}\right). \end{array}$$

The proofs of Propositions 2 and 3 follow by arguments and tools similar to Proposition 1 and are given in Section A.1 for completeness. The new derived information-theoretic conservation laws and decompositions elucidate the relationships between mutual information, directed information, and transfer entropy. Strengthening these connections is crucial, as each measure has distinct tools and applications, thereby enhancing their cross-utilization potential and potentially bridging various applications. While these proposed relations can indeed be derived through existing algebraic manipulations of mutual information, the technical complexity may make these deductions less apparent.

### 3.4. Relating Dependency Structures and Visual Patterns in the InfoMat

We now show that, beyond the derivation of various information decomposition laws, the InfoMat can be used to identify temporal patterns in the data. To this end, we relate various dependence structures to corresponding visual patterns in the InfoMat. We later demonstrate those relations on data via various estimates of the InfoMat (Section 6). To this end, we assume in this section that the sequence $\left(X^{m},Y^{m}\right)$ is an *m*-fold projection of some jointly stationary stochastic process defined over X×Y. As a first example, we note that when the joint process is independent and identically distributed (i.i.d.), every conditional mutual information with i≠j vanishes. The corresponding InfoMat takes the form $I^{X,Y}=I\left(X;Y\right)I_{m}$ with $I_{m}$ being the *m*-dimensional identity matrix.

Next, we focus on a specific case of interest, when the joint process is jointly Markov with some order *k*. In this case, we know that, for $I_{i,j}^{X,Y}$, when both *i* and *j* are larger than *k*, we obtain a similar value of conditional mutual information due to joint stationarity. This implies that within the square block within $I^{X,Y}$ that is determined by the indices {(i,j)|i>k,j>k}, we have a Toeplitz structure (which is a matrix whose constant along its diagonals.). Furthermore, when |i−j|>k, due to the joint Markov nature of the process, we have $I_{i,j}^{X,Y}=0$. This implies a banded structure of the InfoMat, which is determined by a *k*-width ‘Markov band’ outside the main diagonal. This is useful for InfoMat estimation. As we further elaborate in the next sections, estimating the InfoMat may be generally computationally expensive, as we are required to estimate the $m^{2}$ conditional mutual information terms. However, when the joint process is a stationary Markov of order *k*, the number of distinct conditional mutual information terms reduces to O(km). We demonstrate this relation between dependence structures and visual patterns on a simple example in Figure 1. The discussion readily extends to asymmetric Markov orders for the *X* and *Y* processes.

Finally, we note that, as initially observed in Section 3.1, various areas in the InfoMat correspond to different directions of information. Specifically, values in the upper triangular correspond to information flow in the direction $X^{m}\to Y^{m}$, while information flow in the opposite direction is represented by the lower triangular. Therefore, the trace of $I^{X,Y}$ represented the instantaneous exchange of information, also quantifying its operator norm. We believe that further relations and linear algebraic can be unveiled, with task and setting specific structures. In Section 6, we demonstrate these relations on InfoMat estimates from data.

## 4. InfoMat Estimation via Gaussian Mutual Information

Beyond its theoretical merit as a proof visualization tool, we argue that the InfoMat is also effective for the visualization and analysis of dependence structures in time series data. However, the underlying data distribution is often unknown. Even if it is known, the corresponding conditional mutual information terms may not be given in closed form. To this end, to utilize the InfoMat as a visualization tool in real data setting, an estimator is required. In this section, we propose an approximation of $I^{X,Y}$ from samples of the joint distribution $P_{X^{m},Y^{m}}$.

Estimating $I^{X,Y}$ boils down to the estimation of $m^{2}$ conditional mutual information terms. Without assumptions on the data distribution, the complexity and performance of mutual information estimators tend to deteriorate with the length of the conditioned joint history. That is, the bigger (i,j) are, the more samples are required and the worse the performance of the single $I^{X,Y}$ entry is expected to be. To that end, we begin by proposing a data-efficient estimation of mutual information, focusing on an approximation that follows from the Gaussian mutual information term, which is given in closed form. In this case, estimating the entries of $I^{X,Y}$ boils down to the estimation of the corresponding covariance matrices, whose guarantees are well studied, and for which we can use estimators with parametric error rates. We begin by constructing and analyzing the proposed estimator. Then, we analyze its theoretical performance.

### 4.1. Proposed Estimator

Let $\left(X^{n},Y^{n}\right)∼P_{X^{n},Y^{n}}$ be a given dataset from which we want to estimate $I^{X,Y}$. For simplicity, we assume that all variables have zero mean. We begin by using the following representation of conditional mutual information.

**Lemma** **1.**
*Let $\left(X^{m},Y^{m}\right)∼P_{X^{m},Y^{m}}$, and 1≤i,j≤m. Then,*

(12)
$$\begin{array}{cc} I(X_{i};Y_{j}|X^{i-1},Y^{j-1}) & =H\left(X^{i},Y^{j-1}\right)+H\left(X^{i-1},Y^{j}\right)-H\left(X^{i-1},Y^{j-1}\right)-H\left(X^{i},Y^{j}\right). \end{array}$$

*If $\left(X^{m},Y^{m}\right)$ are jointly Gaussian, then*

(13)
$$I\left(X_{i};Y_{j}|X^{i-1},Y^{j-1}\right)=I_{G,i,j}^{X,Y}=≜\frac{1}{2}\log\frac{\left|K_{X^{i},Y^{j-1}}\right|\left|K_{X^{i-1},Y^{j}}\right|}{\left|K_{X^{i-1},Y^{j-1}}\right|\left|K_{X^{i},Y^{j}}\right|},$$

*where $K_{Z}$ is the covariance matrix of $Z∼P_{Z}$, and $|K_{Z}|$ representing its determinant.*


To estimate $I_{i,j}^{X,Y}$ from a dataset $\left(x^{n},y^{n}\right)$ using the Gaussian estimator (Equation 13), we estimate the corresponding sample covariance matrices ${\hat{K}}_{i,j}≜{\hat{K}}_{X^{i},Y^{j}}$, and plug those into (Equation 13). We denote the Gaussian estimator of $I_{i,j}^{X,Y}$ with ${\hat{I}}_{G,i,j}^{X,Y}\left(x^{n},y^{n}\right)$. Finally, a Gaussian estimator of $I^{X,Y}$ is an m×m matrix whose (i,j) entry is given by ${\hat{I}}_{G,i,j}^{X,Y}$. The Gaussian mutual information estimation procedure is summarized in Algorithm 1.
**Algorithm 1** Gaussian InfoMat Estimation**input:** Data $\left(x^{n},y^{n}\right)$, matrix length *m***output:** Gaussian estimate of $I^{X,Y}$Initialize ${\hat{I}}_{G,i,j}^{X,Y}=0$ for (i,j)∈(1,…,m)×(1,…,m)**for** (i,j) in (1,…,m)×(1,…,m) **do**    Divide $(x^{n},y^{n}$) into datasets$${\left({\left(x^{i-1},y^{j-1}\right)}_{l},{\left(x^{i},y^{j-1}\right)}_{l},{\left(x^{i-1},y^{j}\right)}_{l},{\left(x^{i},y^{j}\right)}_{l}\right)}_{l=1}^{N}$$    Calculate sample covariance matrices    Calculate ${\hat{I}}_{G,i,j}^{X,Y}$ via (Equation 13).**return** Estimated InfoMat.

The proposed Gaussian estimator for $I^{X,Y}$ relies on the estimation of covariance matrices. Therefore, it inherits its guarantees from those of log determinants of sample covariance matrices. We assume that the underlying data-generating process is stationary, and obtain the dataset for the estimation of $I_{i,j}^{X,Y}$ by dividing $\left(x^{n},y^{n}\right)$ into the corresponding i×j sequences. We thus have the following

**Proposition** **4**(Gaussian estimator performance guarantees)**.**
*Let $\left(X^{n},Y^{n}\right)$ be a sequence of jointly Gaussian random vectors over $R^{d_{x}+d_{y}}$ and let $d=\max(d_{x},d_{y})$. Then*
*1.* *Bias: $\;\lim_{n\to \infty }E\left[{\hat{I}}_{G,i,j}^{X,Y}\right]=I_{G,i,j}.$**2.* *Variance: $\;\lim_{n\to \infty }\mathrm{Var}\left({\hat{I}}_{G,i,j}^{X,Y}\right)=O\left(\frac{dm^{2}}{n}\right)=O\left(\frac{1}{n}\right).$*

The proof follows from the analysis in [38,39], and the dependence on $m^{2}$ follows from the division of the dataset $\left(x^{n},y^{n}\right)$ into the corresponding subsequences.

We note that, as *m* grows, the performance of the Gaussian estimator deteriorates, as the corresponding conditional mutual information term considers higher dimensional variables. To this end, we propose an alternative dataset acquisition approach—divide $\left(x^{n},y^{n}\right)$ into n−m samples such that the *l*th subsequence for $infomat_{i,j}$ is given by $\left(x_{l}^{l+i},y_{l}^{l+j}\right)$. This provides us with a significantly larger effective dataset when *n* is not bigger than *m* by orders of magnitude. However, the resulting sampled sequences are no longer i.i.d., and therefore, the estimator’s guarantees no longer hold. We refer to such a dataset as a “correlated dataset”, and use it for the visualization of real-world data when data availability is low.

While estimating $I_{G,i,j}^{X,Y}$ is a simpler task, a Gaussian approximation can capture only partial information when the data distribution is far from a joint Gaussian. We propose an upper bound on the error of using the Gaussian approximation.

**Proposition** **5.**
*Let $\left(X,Y,Z\right)∼P_{X,Y,Z}$ and let $\left(P_{X_{G},Y_{G},Z_{G}}\right)$ be the corresponding Gaussian joint distribution with the same moments as $\left(P_{Z,Y,Z}\right)$. We have the following bound:*

(14)
$$\begin{array}{cc} \left|I\left(X;Y|Z\right)-I\left(X_{G};Y_{G}|Z_{G}\right)\right| & \leq \\ & \;D_{\mathrm{KL}}(P_{X,Y|Z}∥P_{X_{G},Y_{G}|Z}|P_{Z})-D_{\mathrm{KL}}(P_{X|Z}⊗P_{Y|Z}∥P_{X_{G}|Z}⊗P_{Y_{G}|Z}\left|P_{Z}\right)\\ & \;+\max_{z\in Z}D_{\mathrm{KL}}\left(P_{X_{G},Y_{G}|Z_{G}=z}∥P_{X_{G}|Z_{G}=z}⊗P_{Y_{G}|Z_{g}=z}\right)d_{\mathrm{TV}}\left(p_{Z},p_{z_{G}}\right) \end{array}$$



The proof of Proposition 5 is given in Section A.2.

### 4.2. InfoMat Estimation for Discrete Datasets

When the data domain is discrete, i.e., $X^{m}$ and $Y^{m}$ are drawn from some finite sets X and Y, respectively, the Gaussian mutual information estimator is no longer valid. In such a case, we propose a plug-in estimator of mutual information. The estimator relies on the entropic factorization in Lemma 1, followed by a standard plug-in estimator for each entropy term. For completeness, we provide more details on the plug-in estimation methodology and demonstrate its application to the InfoMat in Section A.3. In the proposed applications, the plug-in estimator had demonstrated satisfactory results. However, its complexity grows exponentially with the size of conditioned history. In these situations, context tree weighting methodologies [36] can be utilized, to which our approach seamlessly extends.

## 5. Beyond Gaussian—Neural Estimation

Despite its simplicity and data efficiency, the Gaussian mutual information approximation can only fully capture the dependence structure under strong assumptions, which are often violated [40]. To this end, we propose a conditional mutual information estimator that does not require joint Gaussianity. The algorithm relies on the concept of neural estimation [41,42,43,44], which utilizes a variational formula of the measure of interest and optimizes it with neural networks. With the purpose of maintaining the simplicity of the Gaussian method, we turn to a recent scheme that leverages normalizing flows. We utilize the recently proposed diffeomorphic conditional mutual information estimator [39] that leverages a type of network optimization scheme termed MAFs [45]. MAF-based estimators of mutual information map the jointly distributed pair into a corresponding Gaussian pair, such that the learned mapping is a diffeomorphism (which is a differentiable invertible map with a differentiable inverse). In what follows, we introduce the proposed estimator and discuss its performance.

### 5.1. Masked Autoregressive Flows

This section provides a high-level description of MAFs. For an in-depth discussion, we refer the reader to [39,45]. Consider a pair (X,Y) that has a conditional distribution $P_{XY|Z}$ for some random variables *Z*, such that $P_{XY|Z=z}$ is absolutely continuous for any z∈Z. The employed estimator consists of two stages and relies on obtaining a diffeomorphism that maps (X,Y) into a Gaussian pair $\left(X^{\prime },Y^{\prime }\right)$. The second stage consists of calculation of the $I(X^{\prime };Y^{\prime }|Z)$ which has a simple form and will be a proxy for I(X;Y|Z).

To learn a parametric diffeomorphism, we optimize an MAF. MAFs map samples from a (usually simple) base distribution p(U) to samples from an arbitrary target distribution p(X), assuming both are absolutely continuous and defined on $U\subseteq R^{d}$ and $X\subseteq R^{d}$ for some finite d∈N. We denote the MAF with $T_{\theta }$. MAFs are designed such that their Jacobian is a triangular matrix. This implies that its determinant is simply given by the product of its diagonal. This property is crucial for the design as we are interested in representing the likelihood of the target distribution as a function of the likelihood of the base distribution and the partial derivatives along the parametric map $p_{\theta }$. For example, when $p\left(U\right)=\mathrm{Unif}{\left[0,1\right]}^{d}$, the parametric likelihood of *X* under $T_{\theta }$ is given by(15)$$\log p_{\theta }\left(x\right)=\log\left(d\right)+\sum_{i=1}^{d}\log\left|\frac{\partial u_{i}}{\partial x_{i}}\right|,$$
where $\left|\frac{\partial u_{i}}{\partial x_{i}}\right|$ is the Jacobian of $T_{\theta }$. The map $T_{\theta }$ can be realized by neural networks with masked weight matrices and monotone activations [45]. Training an MAF consists of maximum likelihood optimization. Therefore, it is trained using minibatch gradient methods with (Equation 15) serving as the loss for θ. This scheme readily adapts to conditional distributions $p_{\theta }\left(x|z\right)$. The generalization considers a conditioner model $g_{\theta^{\prime }}\left(z\right)$ with parameters $\theta^{\prime }$, whose purpose is to transfer the relevant information in *Z* about *X* into $T_{\theta }$. The conditional MAF is trained similarly to the unconditional MAF, with the slight change that now $x_{i}=f_{\theta }\left(u\right)+g_{\theta^{\prime }}\left(z\right)$. Note that the conditioner model need not to be a diffeomorphism.

### 5.2. Proposed Estimator

Equipped with an MAF model, we can discuss the proposed diffeomorphic conditional mutual information estimator from [39]. We demonstrate the method on an unconditional mutual information term, and then discuss the required modification to introduce conditioning. Recall that our goal is to map the pair (X,Y) into a Gaussian pair $\left(X^{\prime },Y^{\prime }\right)$. We break this task into a concatenation of two stages. First, we will map (X,Y) into a uniform distribution over ${\left[0,1\right]}^{d_{x}+d_{y}}$ by learning a MAF. Then, the uniform distribution will be mapped into a Gaussian distribution using the inverse of the Gaussian cumulative distribution function (CDF) This method is often referred to as generalized inverse transform sampling [46].

To map (X,Y) into a pair of uniform random variables, the learned MAF maps each variable into a mixture of Gaussian CDFs. Specifically, for both *X* and *Y*, we learn a mapping τ: as follows$$\tau \left(x_{i},h_{i}\right)=\sum_{j=1}^{k}w_{i,j}\left(h_{i}\right)\Phi \left(x_{i};\mu_{i,j}\left(h_{i}\right),\sigma_{i,j}^{2}\left(h_{i}\right)\right),$$
where Φ is the Gaussian CDF with mean μ and variance $\sigma^{2}$, ${\left(w_{i,j}\right)}_{i,j}$ are the model parameters, and $h_{i}$ is some parametric autoregressive summary map, i.e., $h_{i}=h_{i}\left(x_{<i}\right)$. As explained in [39], the reasoning behind this design choice is that $\partial \tau /\partial x_{i}$ is a Gaussian mixture model, which is known to be a universal approximator of probability distributions. Finally, the decomposition of τ and the inverse Gaussian CDF yields a parametric model that maps an arbitrary random variable into a Gaussian variable. Thus, when applied to *X* and *Y*, we result with a pair of Gaussian variables $X^{\prime }$ and $Y^{\prime }$.

Having mapped (X,Y), which are distributed according to the conditional distribution $P_{XY|Z}$ into the Gaussian pair, we can use the following result on conditional mutual information invariance.

**Proposition** **6**(Conditional mutual information invariance)**.**
*Let $\left(X,Y,Z\right)∼P_{X,Y,Z}$ and denote by $\left(X^{\prime },Y^{\prime }\right)=\left(f_{\theta }\left(X,Z\right),g_{\phi }\left(Y,Z\right)\right)$, where $f_{\theta }$ and $g_{\phi }$ are conditional diffeomorphisms. Then,*(16)$$I\left(X;Y|Z\right)=I\left(X^{\prime };Y^{\prime }|Z\right)$$

Proposition 6 is a slight modification of ([39] (Lemma 2)). The existence of optimal MAFs is guaranteed by the universal approximation properties of normalizing flows [47]. Finally, the Gaussian mutual information term is calculated from sample covariance matrices, as elaborated in Section 4. The MAF-based scheme is depicted in Figure 2, and the algorithm steps are summarized in Algorithm 2.

To estimate $I^{X,Y}$ from a given dataset, we apply Algorithm 2 to each coordinate pair (i,j). The data are split in a similar fashion to Algorithm 1, but due to the parametric nature of the estimator, we optimize using iterative minibatch-gradient descent. In the training phase, we optimize all DMI models in parallel through the optimization of the corresponding maximum-likelihood loss (Equation 15) for a fixed number of epochs. When the training concludes, for each entry, $I_{i,j}^{X,Y}$, we feed the corresponding dataset through the optimized MAFs and estimate the sample covariance matrices, from which we calculate the mutual information term. Neural network optimization is considerably expensive. However, ref. [39] shows that the DMI outperforms existing conditional mutual information estimators in terms of sample requirements. The proposed method boils down to the optimization of $m^{2}$ MAF models, which may be computationally expensive. We believe that this complexity can be alleviated by incorporating recurrent architectures or attention models. However, this investigation is out of the scope of this work.
**Algorithm 2** Neural InfoMat Estimation**input:** Data $\left(x^{n},y^{n}\right)$, matrix length *m*, number of epochs *N*.**output:** Neural estimate of $I^{X,Y}$Initialize MAF parameters $\theta_{i,j}$ for (i,j)∈(1,…,m)×(1,…,m)**for** (i,j) in (1,…,m)×(1,…,m) **do**    Divide $(x^{n},y^{n}$) into datasets$${\left({\left(x^{i-1},y^{j-1}\right)}_{l},{\left(x^{i},y^{j-1}\right)}_{l},{\left(x^{i-1},y^{j}\right)}_{l},{\left(x^{i},y^{j}\right)}_{l}\right)}_{l=1}^{N}$$    Optimize $T_{\theta_{i,j}}$ for *N* epoch via maximum likelihood (Equation 15)    Calculate ${\hat{I}}_{G,i,j}^{X,Y}$ via (Equation 13) on $T_{\theta_{i,j}}$ to the sample set.**return** Estimated InfoMat.

**Remark** **1**(Performance of normalizing flows)**.**
*MAFs are a recent promising method for the estimation of (conditional) mutual information using the expressive power of neural network, and were shown to outperform previous methodologies in an array of experiments [39]. However, using the MAF method separately on X and Y yields a pair of Gaussian variables, which are not guaranteed to be jointly Gaussian [48]. Consequently, the estimated mutual information (through the Gaussian formula) is, in general, a lower bound of I(X;Y). Nonetheless, the proposed method showed good empirical performance in considered tasks, as we show in Section 6.*

**Remark** **2**(Computational Complexity and Scalability)**.**
*Estimating the m×m InfoMat, where each entry represents a conditional mutual information term, poses significant computational challenges. In the neural estimation approach, each of the $m^{2}$ entries is estimated via a separate neural network, leading to an overall computational complexity of $O(Tm^{2})$, where T is the complexity of estimating a single conditional mutual information term. This quickly becomes prohibitive as m increases, both in terms of training time and memory usage. This complexity can potentially be reduced by incorporating weight or state sharing through recurrent architectures, which we aim to explore in future work. The current implementation offers two estimation options: the first is the neural estimator, which provides a more accurate estimation when ample data are available, albeit with increased computational overhead. The second is the Gaussian mutual information Estimator, which relies on covariance matrix estimation and offers a more computationally efficient alternative that performs well in low-data regimes or for moderate values of m.*

## 6. Visualization of Information Transfer

In this section, we demonstrate the utility of the InfoMat as a visualization tool for sequential data. We demonstrate how one can use the relations between information measures and InfoMat entry subsets (Section 3) to deduce temporal interactions in sequential data. We show that the InfoMat provides a more informative mode of information compared to existing measures, and propose measurements that can be coupled with the InfoMat estimate to deduce relationships in the data. Throughout, we adopt the interpretation that higher directed information in a certain direction implies a higher causal effect [36,43]. We analyze the InfoMat through its heatmap representation. Specifically, the heatmap *X*-axis corresponds to the *X*-process time index and *Y*-axis corresponds to the *Y*-process time index, as per the InfoMat definition (Equation 6). An implementation of considered experiments can be found at https://github.com/DorTsur/infomat (accessed on 22 March 2025).

### 6.1. Synthetic Data—Gaussian Processes

We begin with a sequential Gaussian process, which allows us to clearly present the relations between various dependence structures and the resulting InfoMat structure. Specifically, we consider the following joint Gaussian autoregressive (AR) process$$\begin{array}{cc} X_{t} & =\sum_{k=0}^{{\bar{k}}_{x}}\alpha_{k}^{X}X_{t-k}+\alpha_{k}^{Y}Y_{i-k}+N_{t}^{X},\;t\in N\\ Y_{t} & =\sum_{k=0}^{{\bar{k}}_{y}}\beta_{k}^{X}X_{t-k}+\beta_{k}^{Y}Y_{t-k}+N_{t}^{Y}, \end{array}$$
where $N_{t}^{Y}$ and $N_{t}^{Y}$ are samples of a centered i.i.d. Gaussian processes with covariance matrices $K_{N_{x}}$ and $K_{N_{y}}$, respectively. By controlling the values for the AR model parameters ${\left({\bar{k}}_{x},{\bar{k}}_{y},\alpha_{k}^{X},\alpha_{k}^{Y},\beta_{k}^{X},\beta_{k}^{Y}\right)}_{k=1}^{m}$, we induce various dependence structures on the sequence $\left(X^{m},Y^{m}\right)$, which we then visualize via $I^{X,Y}$. All visualizations in this subsection are obtained via Algorithm 1, with $n\approx 10^{5}$ samples, which as. We assume that the samples are given from the stationary distribution by omitting the first $\max({\bar{k}}_{x},{\bar{k}}_{y})$ samples.).

We begin with a simple i.i.d. setting by taking $\beta_{k}^{X}=\gamma$ for γ∈(0,1) and nullifying the rest of the parameters. In this case, $X_{i}⊥\;⊥Y_{j}$ for i≠j. Thus, all shared information is instantaneous, implying that we should expect a diagonal InfoMat. As seen in Figure 3a, this is indeed the case. The corresponding InfoMat captures the dependence structure, resulting in a diagonal matrix.

Next, denote ${\bar{x}}_{x}={\bar{k}}_{y}=\bar{k}$. We increase the values of AR weights for $\bar{k}>0$, inducing dependence in the history of the joint process. We consider a symmetric dependence structure and set $\alpha_{k}^{X}=\alpha_{k}^{Y}=\beta_{k}^{X}=\beta_{k}^{Y}=0.3$ for $j\in \{0,\ldots ,\bar{k}\}$ considering several values of $\bar{k}$. The value of $\bar{k}$ can be interpreted as how far into the past the dependence of the present terms extends. As shown in Figure 3b,c, the larger $\bar{k}$ is, the bigger the ‘information band’ around the diagonal, whose width depends on the value of $\bar{k}$. Consequently, we may deduce that the farther away we reside on the off-diagonal, the farther in history we observe dependence.

Finally, we demonstrate how more complex temporal structures can be captured via the InfoMat. Specifically, we take two cases in which the ARMA parameters are time-varying, leading to nontrivial temporal relations between the processes. The first considers that ARMA parameters vary over time, i.e., the amount of parameters which are not nullified depends on the (i,j) value. As seen in Figure 4a, this results in a decay in the aforementioned ‘dependence band’. Second, we demonstrate a case of pure unidirectional information transfer in a single direction by introducing a time delay in the parameters and a significant difference between their values. As shown in Figure 4b, the information transfer in dense in the lower triangular, which correspond to the DI term $I(Y^{m}\to X^{m})$ (see Section 3). We conclude that the InfoMat successfully captures nontrivial temporal dependence structures.

### 6.2. Expressiveness of Neural Estimation

As previously discussed, the efficient InfoMat estimation through Gaussian conditional mutual information estimation comes at the cost of a mismatch when the relations are, e.g., nonlinear. Herein, we demonstrate the utility of neural estimation (Section 5) to the InfoMat capabilities by comparing its performance with the Gaussian mutual information estimator. We take an i.i.d. jointly Gaussian dataset with correlation coefficient ρ=0.9. We then apply a cyclic shift of T<m to the samples $Y^{m}$ within each *m*-length sequence, effectively resulting in a time-shift dependence structure. We introduce nonlinearity by considering the mapping $X_{i}\mapsto \log\left(X_{i}\right)$ and $Y_{i}\mapsto Y_{i}^{3}$. Such mappings are invertible and therefore the overall mutual information should be preserved.

As visualized in Figure 5, the neural estimator successfully recovers the correct structure in the data, while the Gaussian mutual information estimator fails to provide a meaningful visualization in the given setting. As the mappings are invertible, the resulting mutual information is 0.83 [nats] on nonzero entries, which are approximately the corresponding values in the neural estimator of $I^{X,Y}$. However, this accuracy comes at the cost of training $m^{2}$ neural nets, which is significantly slower than calculating sample covariance matrices. Consequently, we argue that one should consider neural estimation when the functional nature of the data are complex, and consider the Gaussian estimator when *m* is large, or when simple, initial results may be of need.

### 6.3. Visualizing Information Flow in Physiological Data

Directed information and transfer entropy have been previously addressed as measures of causal effect between two interacting processes [36,49]. To that end, they have been used to quantify and compare flows of information in stochastic systems. We demonstrate this paradigm with the InfoMat, while showing its applicability to visualize dynamics in real-world datasets. We consider the Apnea dataset from Santa Fe Time Series Competition (https://physionet.org/content/santa-fe/1.0.0/ (accessed on 22 March 2025)) [50,51]. The Apnea dataset is common benchmark for the evaluation of transfer entropy estimation. It is a sequential dataset which consists of measures of heart rate and chest volume (representing respiration force). The Apnea dataset was previously addressed in the literature as a case study to understand the relation between sequential information measures and causal effect. We estimate the InfoMat on the Apnea dataset, denoting the interacting processes at hand being ${\left(X_{t}\right)}_{t\in N}=\mathrm{Heart}$ and ${\left(Y\right)}_{t\in N}=\mathrm{Breath}$. The visualization of the estimated InfoMat is given in Figure 6.

It was shown in [37,49] that the transfer entropy in the direction Breath→Heart is higher than the transfer entropy measures in the other direction. By calculating the directed information in each direction, we recover the same conclusion on the relationship in the data. Specifically, we have$$\hat{I}\left(D\circ X^{n}\to Y^{n}\right)=0.14<0.41=\hat{I}\left(D\circ Y^{n}\to X^{n}\right),\;{\hat{I}}_{inst}\left(X^{n},Y^{n}\right)=0.034.$$
This calculation implies that the causal effect in the Breath→Heart direction is the prominent one, which is in agreement with the previous literature. This conclusion provides another validation of the InfoMat method consistency.

Beyond its agreement with known previous results, the InfoMat provides a more informative observation of the exchange of control between the two measures. For example, we observe that most of the information transfer occurs in the first upper subdiagonal, which corresponds to $I(X_{i-1};Y_{i}|X^{i-2},Y^{i-1})$, i.e., most information is transferred from the previous time step to the next one. In the opposite direction, we have a significantly smaller information transfer. Surprisingly, information is transferred from the steps further in the process past, i.e., the effect is from $Y_{t-4}^{t-2}$ to $X_{t}$. These results further motivate the use of the InfoMat as a visual tool for the task of exploratory data analysis, we apply it to real-world data.

### 6.4. Visualizing Coding Schemes Effect

As a final application, we demonstrate an application of the InfoMat to analyze information flow in digital communication schemes. In this case, $X^{m}$ and $Y^{m}$ represent the input and output of some communication channel with memory. The joint distribution of $\left(X^{m},Y^{m}\right)$ is determined by the channel transition kernel input distribution. The purpose of the encoder, which generates the sequence $X^{m}$ according to some causally conditional law $P\left(X^{m}∥Y^{m-1}\right)≜\prod_{i=1}^{m}P\left(X_{i}|Y^{i-1}X^{i-1}\right)$ [26], is to maximize the downstream directed information. Formally, under mild assumptions, the *feedback* capacity is characterized by the following optimization problem [7](17)$$C=\lim_{n\to \infty }\underset{P(X^{n}∥Y^{n-1})}{\sup}\frac{1}{n}I\left(X^{n}\to Y^{n}\right).$$
where C is termed the ‘feedback capacity’ of the communication channel, which is determined by the causal conditional law $P\left(Y^{m}∥X^{m}\right)≜\prod_{i=1}^{m}P\left(Y_{i}|Y^{i-1}X^{i}\right)$. Solving the channel capacity optimization (Equation 17) provides the user with the value of maxima achievable rate of reliable communications, coupled with a coding scheme that achieves this rate under block length asymptotics. In what follows, we visualize the InfoMat for several channels, under both a channel oblivious coding scheme and the capacity-achieving coding scheme. As the data for this application are discrete, we utilize a simple plug-in estimator for the considered conditional mutual information terms. For completeness, we provide a characterization and analysis of the plug-in entropy estimator in Section A.3.

#### 6.4.1. Ising Channel

As a first example, we consider the Ising channel [52], which is a popular example of a communication channel with memory, which adheres to the famous Ising model. In this case, the channel law is defined according to the relation$$Y_{t}=\left\{\begin{array}{c} X_{t},\;w.p.\;0.5\\ X_{t-1},\;w.p.\;0.5 \end{array}\right.$$
The feedback capacity-achieving coding scheme was obtained in [53] by representing the Ising channel is a finite state channel [7], which allowed for a dynamic programming approximation of the corresponding optimization.

We estimate the InfoMat in the Ising channel under two coding schemes. The first, which we refer to as the oblivious scheme, sends $X^{m}\overset{i.i.d.}{∼}\mathrm{Ber}\left(1/2\right)$ independently of the channel outputs. (Figure 7a). The second generates $X^{m}$ according to [53] (Figure 7b). We note that the oblivious scheme generates an InfoMat with most of its information in the main diagonal and a small residue in the off-diagonal. The diagonal information follows from the i.i.d. scheme, and it is constant along all time steps. The nonzero off-diagonal entries are due to event $Y_{t}=X_{t-1}$ which injects memory through the channel transition kernel. When we consider the feedback capacity-achieving scheme from [53], we result with a non-constant pattern of information flow. Specifically, most of the information is sent along the off diagonal, i.e., most of the information is sent through the effect of past channel inputs. Additionally, we note that the amount of conveyed information is non-constant. This is a result of the underlying finite-state machine that defines the evolution of $X^{m}$ according to past inputs, outputs and states ([53] Figure 5).

Finally, we calculate the normalized directed information for each scheme, as it serves as the proxy for the information rate in the channel. We have$$\frac{1}{m}{\hat{I}}_{i.i.d.}\left(X^{m}\to Y^{m}\right)\approx 0.45,\;\frac{1}{m}{\hat{I}}_{\mathrm{opt}}\left(X^{m}\to Y^{m}\right)\approx 0.56$$
We note that, as expected, the capacity-achieving scheme yields a significantly greater normalized directed information. The resulting quantity is very close to the theoretical capacity values (0.575). We conjecture that the mismatch results from the plug-in estimation error. Finally, we note that, in contrast to the i.i.d. scheme, the optimal scheme generates information in the direction Y→X as well, which quantifies the usage of feedback in the scheme.

#### 6.4.2. Trapdoor Channel

Next, we visualize the effect of the coding strategy on the transfer of information in the Trapdoor channel. The trapdoor channel is an example of a binary finite-state channel whose state and output evolve according to the relation$$Y_{t}=\left\{\begin{array}{cc} Z_{1/2}\left(X_{t}\right), & \mathrm{if}\;S_{t-1}=0\\ S_{1/2}\left(X_{t}\right), & \mathrm{otherwise} \end{array}\right.,\;S_{t}=S_{t-1}⊕X_{t}⊕Y_{t},$$
where $Z_{1/2}$ and $S_{1/2}$ denote the Z- and S-channels with a probability of 1/2. The output of a $Z_{p}$$\left(S_{p}\right)$ channel equals its input when the latter is 0 (1) and distributes according to Ber(p) otherwise. These channels are fundamental and have been extensively investigated in the literature [4,54]. Again, we consider the channel oblivious coding scheme and the optimal coding scheme from [7], as can be seen in Figure 8. We note that the highest amount of conveyed information is in the beginning of transmission. Notably, the information transfer under the optimal Trapdoor scheme is more uniform than the optimal Ising coding scheme. In this case, the estimated normalized directed information is$$\frac{1}{m}{\hat{I}}_{i.i.d.}\left(X^{m}\to Y^{m}\right)\approx 0.441,\;\frac{1}{m}{\hat{I}}_{\mathrm{opt}}\left(X^{m}\to Y^{m}\right)\approx 0.663$$
Again, we note that the optimal coding strategy induces information transfer in the backward direction Y→X due to the incorporation of feedback into the input distribution.

## 7. Conclusions

In this work, we developed the InfoMat, a matrix representation of information exchange. We showed the utility of the InfoMat for the visual proofs of information conservation laws via matrix coloring arguments, which allowed us to expand the existing set of decompositions for information measures. Then, we proposed several estimators of the InfoMat, which were studied both theoretically and empirically. Equipped with the InfoMat estimators, we presented several applications of the InfoMat as a visualization tool to analyze the dependence structures and information transfer in sequential datasets in various settings. For future work, we aimed to develop a computationally efficient neural estimator of the InfoMat using weight sharing, sequential architectures [55], and slicing techniques [56]. Given this work’s simplicity and the popularity of information measures, the InfoMat can serve as an effective tool for data exploration in sequential data analysis pipelines. Furthermore, we believe that the InfoMat can be highly useful for a myriad of contemporary research fields that involve time series. Such fields encompass empowerment [14], which characterizes robust sequential decision-making via information theory, and causal inference [16], in which information theory has been shown to be beneficial. Finally, we aim to investigate the multivariate extensions of the InfoMat, as it is central to contemporary setting. This extension can be obtained by constructing higher-dimensional information tensors that capture conditional mutual information among multiple signals. while this extension can uncover rich patterns of inter-dependencies in complex datasets, it also introduces new challenges in computational scalability and visualization clarity.

## Figures and Tables

**Figure 1 entropy-27-00357-f001:**
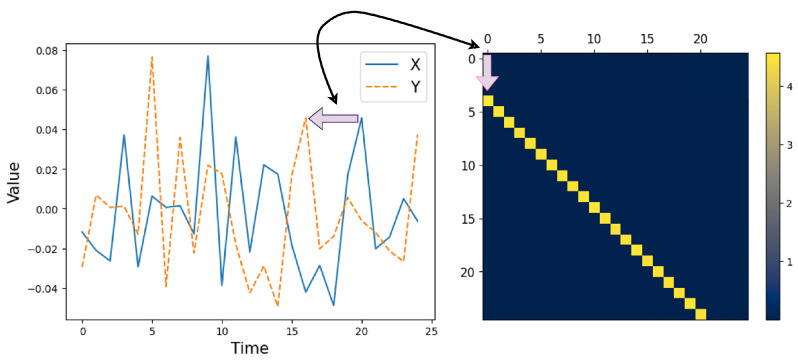
Visualizing temporal dependencies in the InfoMat. We take a simple Gaussian process, where $X_{t}\overset{i.i.d.}{∼}N\left(0,1\right)$ and $Y_{t}=X_{t+4}+Z_{t}$ with $Z_{t}\overset{i.i.d.}{∼}N\left(0,0.1\right)$. In this case, the data show that (**left**) $Y_{t}$ follows $X_{t}$ with a delay of 4 time-steps. This is also visible in the InfoMat by a shift of 4 indices of the InfoMat Heatmap representation (**right**). The acquisition of the InfoMat Heatmap representation is explained in Section 6.

**Figure 2 entropy-27-00357-f002:**
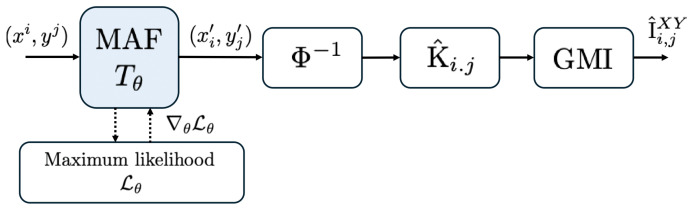
Neural estimation model. Dashed line represents maximum-likelihood (ML) training phase, while the filled lines account for the inference (mutual information calculation) phase.

**Figure 3 entropy-27-00357-f003:**
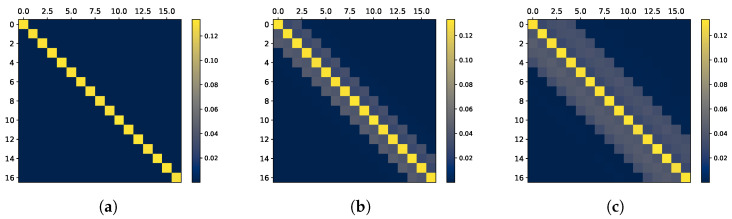
Visualization of history dependence in ARMA Gaussian process with various parameter settings. The bigger the value of $\bar{k}$ is, the bigger the information band around the instantaneous information, represented by the diagonal. (**a**) Gaussian i.i.d. ($\bar{k}=0$). (**b**) Gaussian AR, $\bar{k}=2$. (**c**) Gaussian AR, $\bar{k}=4$.

**Figure 4 entropy-27-00357-f004:**
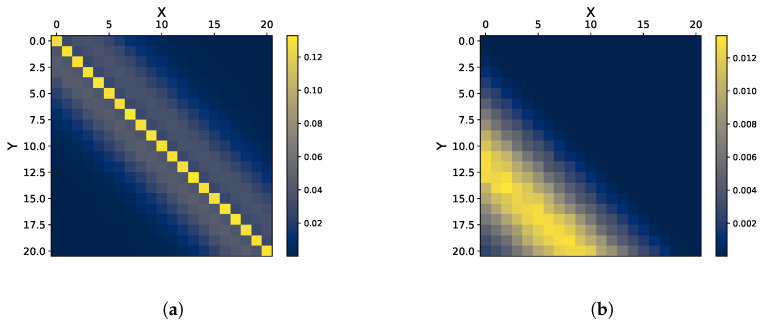
Visualization of complex dependence structures in the Gaussian AR setting. (**a**) Decaying Gaussian AR, γ=0.5. (**b**) One-sided information transfer.

**Figure 5 entropy-27-00357-f005:**
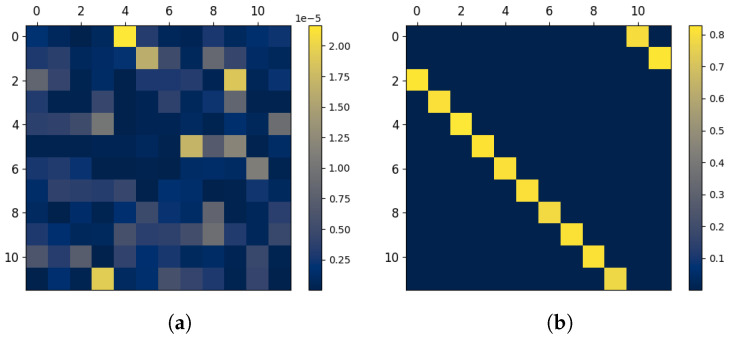
Estimated InfoMat under nonlinearities and cyclic shift. (**a**) Gaussian mutual information. (**b**) Neural estimator.

**Figure 6 entropy-27-00357-f006:**
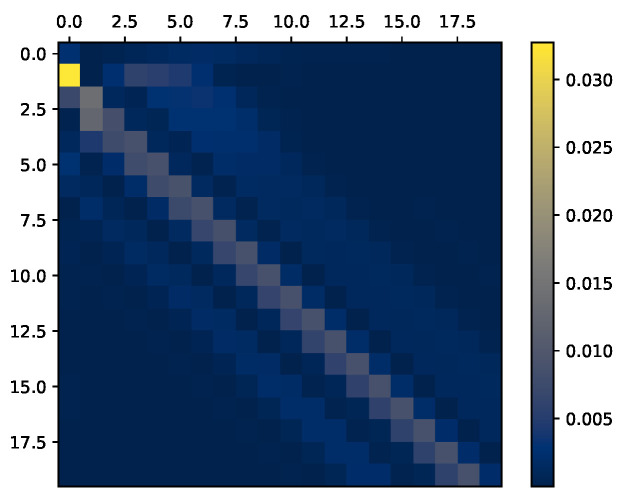
Physiological data. Larger effect in observed in the ‘breath’→‘heart’ direction.

**Figure 7 entropy-27-00357-f007:**
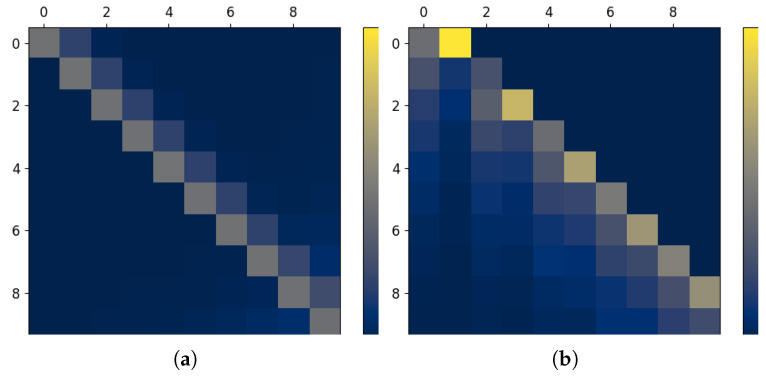
Visualization of information transfer in the Ising channel under various coding schemes. (**a**) Oblivious coding scheme. (**b**) Optimal coding scheme.

**Figure 8 entropy-27-00357-f008:**
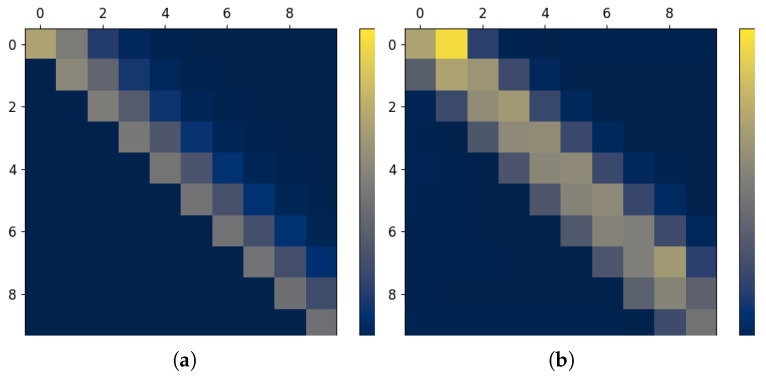
Visualization of information transfer in the Trapdoor channel under various coding schemes. (**a**) Oblivious coding scheme. (**b**) Optimal coding scheme.

**Table 1 entropy-27-00357-t001:** Visual shapes of dependence patterns in $I^{X,Y}$.

Information measure	Visual pattern in $I^{X,Y}$
$I(X^{m}\to Y^{m})$	Upper triangular **with diagonal**
$I(D^{k}\circ X^{m}\to Y^{m})$	Upper triangular, side (m−k)
$T_{i+1}^{X\to Y}\left(i,i\right)$	Col. in row i+1 with length *i*

## Data Availability

All data and experiments are available at the publig GirHub repository at https://github.com/DorTsur/infomat (accessed on 22 March 2025).

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
