# Peer review of "InfoMat: Leveraging Information Theory to Visualize and Understand Sequential Dataâ€"

_entropy, 2025, doi:10.3390/e27040357_

Round 1
Reviewer 1 Report
Comments and Suggestions for Authors
Report on “InfoMat: Leveraging Information Theory to Visualize and Understand Sequential Data” submitted for publication in Entropy.
This work puts forward a visualization tool (a matrix) called InfoMat, based on Information Theory, and which is used to quantify information flow in sequential data. The authors also show this new tool is readily able to recover results of conservation of information and reveal new ones. Alongside with this new tool, the authors propose two estimators and test them, showing very good and promising results for the neural estimator when applied to test settings.
I believe the work is interesting, theoretically sound, and good as it is and I already suggest its acceptance for publication.
I would only like to make some minor suggestions to the authors, which they are free to implement or not.
General comments:
- My only suggestion to the authors would concern the lack of citations of applications of information theory to analyze sequences (in line 34). Since the paper was sent for publication in Entropy, I believe the authors could establish an easy connection to the physics literature by citing measures/metrics/statistics (related to information theory) proposed in the context of physics/nonlinear dynamics such as: sample entropy, approximate entropy, permutation entropy, dispersion entropy, and others. I make these suggestions somewhat loosely so the authors may better judge which of these are more in line with their work.
Typos and more practical questions regarding the text:
- line 44: “only when vectors entries” -> “only when vector entries”
- line 89: The authors mention the variable $P$ without previously defining it
- line 93: “We The entropy” -> “The entropy”
- line 239: “an approximation the follows” -> “an approximation that follows”
- line 266: “follows fro the division” -> “follows from the division”
- line 290: “its complexity is grows exponentially” -> “its complexity grows exponentially”
- line 371: “mutual ifnformation” -> “mutual information”
- line 393: “are samples a of centered” -> “are samples of centered”
- line 408: “Consequently, We may” -> “Consequently, we may”
- line 418: “figure 4b” -> “Figure 4b”
- Legend of Fig. 4: “Visualization Complex” -> “Visualization of complex”
- line 442: “entropy has been” -> “entropy have been”
- line 478: “coding schemes” -> “coding scheme”
Author Response
We would like to deeply thank the reviewer for their positive feedback and important comments. Below are our responses to the reviewer's two main comments.
"Typos and more practical questions regarding the text: ..."
Thank you for spotting the typos. The spotted typos were fixed and additional proofreads had been done to the manuscript to eliminate any possible additional typos and errors.
"My only suggestion to the authors would concern the lack of citations of applications of information theory to analyze sequences (in line 34). Since the paper was sent for publication in Entropy, I believe the authors could establish an easy connection to the physics literature by citing measures/metrics/statistics (related to information theory) proposed in the context of physics/nonlinear dynamics such as: sample entropy, approximate entropy, permutation entropy, dispersion entropy, and others. I make these suggestions somewhat loosely so the authors may better judge which of these are more in line with their work."
Thank you for this suggestion. we have added a discussion on the mentioned measures (sample entropy, permutation entropy and dispersion entropy) in the background section, as those are fundamental measures for quantification of information in time-series for which we weren't aware of.
Please see the revised manuscript with changes highlighted in blue for specific information on the added discussion.
Reviewer 2 Report
Comments and Suggestions for Authors
Thanks for sharing your work!
I like the idea of using a matrix-based visualization to represent sequential information flow, as it provides an intuitive way to analyze complex dependencies over time.
Below are a few points that might help strengthen the paper further:
- In my opinion, InfoMat is essentially a matrix of conditional mutual information terms, which might feel more like a reorganized presentation of well-known components rather than an entirely new approach. Hence, at this stage, InfoMat appears to be a neat visualization of time-lagged conditional mutual information. Can you further emphasize why this matrix form fundamentally aids analysis?
- Obviously, estimating an m \times m matrix of conditional mutual information can become expensive, especially for large m. Moreover, when neural estimators are used for each entry, the overhead can be huge. I’d appreciate either (a) a more efficient algorithm (maybe using some recurrent or weight-sharing scheme) or (b) an honest discussion of computational feasibility for large m.
- It would help to include a brief comparison of InfoMat’s insights vs. classic approaches like partial correlations, cross-correlation matrices, Granger causality, or standard directed information metrics on entire sequences. A simple experiment to illustrate “here’s what other tools give, here’s how InfoMat reveals something else’’ would be quite compelling. I noticed that the authors have published several papers in this line of work, so you have likely discussed or compared these methods in previous publications. If so, please briefly refer to those discussions.
- You focus on bivariate sequences; however, real-world applications often involve multiple signals, not just two. Can you provide some guidance (even if still conceptual) on how InfoMat might extend to multi-variate sequences?
- Although it may not be feasible for the current revision time, as a reader, I would expect to see extensive tests on synthetic datasets with known ground truth. This should include running times, memory usage, and performance in low-sample regimes to demonstrate practical viability. I also find a lack of estimation reliability analysis in your manuscript.
- The communication-channel and apnea examples are nice, but it’d be great to see a bit more real-world complexity. For instance, do we truly discover insights about data structure or dynamics that are missed by simpler measures? Show that advantage to the reader.
- Since InfoMat can grow quite large, interpreting so many conditional mutual information entries might be daunting. I think you also need to work on user-friendly interpretability.
The core idea of a matrix-based visualization for sequential information flow is interesting, and I suggest accepting this manuscript with the above minor revisions.

Author Response
We deeply thank the reviewer for their great feeback and comments. Please see the attached file for our detailes responses. We have submitted a revised version of the manuscript with changes highlighted in blue.
